# Observation of strong nonlinear interactions in parametric down-conversion of X-rays into ultraviolet radiation

S. Sofer[1,4], O. Sefi[1,4], E. Strizhevsky[1], H. Aknin[1], S.P. Collins[2], G. Nisbet [2], B. Detlefs [3], Ch.J. Sahle [3] & S. Shwartz [1]*

Nonlinear interactions between X-rays and long wavelength radiation can be used as a powerful atomic-scale probe for light-matter interactions and for properties of valence electrons. However, reported X-ray nonlinear effects were small and their observations required tremendous efforts. Here we report the observation of strong nonlinearities in parametric down-conversion (PDC) of X-rays to long wavelength radiation in gallium arsenide and lithium niobate crystals, with efficiencies about 4 orders of magnitude stronger than the efficiencies measured in any material studied before. Furthermore, we show that the efficiency in the ferroelectric phase of strontium barium niobite is two orders of magnitude stronger than in its paraelectric phase. This observation suggests that the lack of inversion symmetry is the origin for the strong observed nonlinearity. Additionally, we demonstrate the ability to use the effect for the investigation of the spectral response of non-centrosymmetric materials at wavelengths ranging from infrared to soft X-rays.

[1] Physics Department and Institute of Nanotechnology, Bar-Ilan University, 52900 Ramat Gan, Israel. [2] Diamond Light Source, Harwell Science and Innovation Campus, Didcot OX11 0DE, UK. [3] ESRF–The European Synchrotron, CS 40220, 38043 Grenoble Cedex 9, France. [4]These authors contributed equally: S. Sofer, O. Sefi. *email: sharon.shwartz@biu.ac.il

Optics and photonics is a major field of research that is important for fundamental sciences and has led to many practical applications and inventions. This field, which covers a broad range of the electromagnetic spectrum is, to a large extent, based on light-matter interactions. It is clear that a better understanding of the light-matter interactions is crucial for further developing of photonics-based research and technologies. Indeed, the approaches and techniques aiming at improving the knowledge of light-matter interactions are diverse and attract a great deal of attention[1]. However, most approaches utilize long wavelength radiation, thus cannot probe microscopic scale information, such as the valence electron redistribution in response to the application of external electric field or illumination. This is due to the fundamental limit of resolution of waves, which limits the resolution to about the half of the wavelength. Although X-ray-based techniques are capable of probing atomic-scale structures, they interact mainly with core electrons, thus provide very limited information on valence electrons and on their atomic-scale interactions with long wavelength radiation[2]. There are several methods that utilize Bragg diffraction to reconstruct the valence charge density by subtracting the theoretical contribution of the core electrons from the measured intensity[3,4]. However, those methods are very limited[3] and the experimental results exhibit large variance and are often not reproducible[4]. Moreover, those methods cannot provide any spectroscopic information about the optical response of valence electrons.

About half a century ago, Freund[5,6] proposed a method to measure the microscopic properties of valence electrons in solids by using nonlinear wave mixing of X-rays and long wavelength radiation such as UV and visible. In essence, the effect can be viewed as X-ray scattering from optically modulated valence electrons, as is shown schematically in Fig. 1a, hence in this process the X-rays probe the variation in the state of the valence electrons and provide the access to the microscopic world. The advent of new X-ray sources such as the third-generation synchrotrons and more recently free-electron lasers has led to a substantial progress in the field. Previous works that studied X-ray and optical wave mixing exploited the effect of sum-frequency generation and PDC[5–16]. The effect of difference frequency generation of two X-ray pulses was studied theoretically as a probe for microscopic properties at the atomic scale[17,18]. Moreover, the effect of two photon absorption has been used recently for nonlinear spectroscopy[19]. The nonlinearity in X-ray and long wavelength mixing was discussed by numerous theoretical works[20–25]. However, to date, nonlinear interactions between X-rays and long wavelength radiation have been observed only in simple crystals such as diamond and silicon[8–15]. The typical efficiencies and the signal-to-noise ratios that have been reported are very small and are barely sufficient for the measurements. The experimental results have been interpreted by using simple classical or semiclassical models for the nonlinearity with reasonable agreements where the long wavelengths are either in the range where the crystals are transparent[11,14] or in the extreme ultraviolet regime with only one isolated resonance[8–10,12,13,15].

In the effect of PDC of X-rays into longer wavelength radiation, an input X-ray pump beam interacts with the vacuum fluctuations in the nonlinear crystal to generate photon pairs at lower frequencies. Energy conservation implies that the wavelength of one of the generated photons is in the X-ray regime (denoted as signal) and the wavelength of the second photon is in the UV or visible range (denoted as idler) where the sum of the frequencies of the generated photons is equal to the frequency of the input X-ray photon, namely $\omega_p = \omega_s + \omega_i$. Due to the high absorption in the UV range, the UV photons are completely absorbed and only the X-ray photon can be detected. However, since the photons are always generated in pairs, the rate of the X-rays depends also on the optical properties of the material in the UV range, hence the measurement of X-rays is sufficient to retrieve the information that can be probed by the UV photons. The selection of the wavelengths of the generated beams is done by using the requirement for momentum conservation (phase matching) that imposes a relation, which depends on the refractive indices of the material, between the propagation angles of the beams and their photon energies. Since the wavelengths of the X-ray photons are on the order of the distances between atomic planes, we utilize the reciprocal lattice vectors to achieve phase matching, which is given by $\mathbf{k}_p + \mathbf{G} = \mathbf{k}_s + \mathbf{k}_i$ where $\mathbf{k}_p$, $\mathbf{k}_s$, and $\mathbf{k}_i$ are the wavevectors of the pump, signal, and idler beams, respectively and $\mathbf{G}$ is the reciprocal lattice vector.

In this work we report on the observation of prominent large nonlinear effects in the non-centrosymmetric crystals gallium arsenide (GaAs) and lithium niobate (LiNbO$_3$), which are stronger by orders of magnitude from the background. Our measurements of the effect of PDC of X-rays into the UV and optical regimes in these crystals exhibit efficiencies that are about four orders of magnitude stronger than the efficiencies measured in any crystal, in which X-ray PDC has been observed so far. This is in sharp contrast to the prediction of the existing theories. Our results indicate an unrevealed underlying physical mechanism that is responsible for the strong nonlinearity we observed. As a first step in the direction of understating the observed strong nonlinearity we explore the lack of inversion symmetry as a possible physical mechanism. In addition, we demonstrate the ability to perform spectroscopic measurements in a very broad wavelength range that enable the retrieval of information on the band structure, density of states, and atomic resonances in the crystals. As such, our work opens exciting possibilities for future research of nonlinear X-ray optics in complex non-centrosymmetric materials and surfaces, and for the development of advanced spectroscopy techniques that rely on these effects.

## Results

**Experimental setup.** We conducted the experiments described in this article on beamline ID20 of the European Synchrotron Radiation Facility and on beamline I16 of the Diamond Light Source[26]. The schematic of the experimental setup is shown in Fig. 1b. The setup is very simple and relies on a standard diffractometer. We used a crystal analyzer to select the photon energy of the detected outgoing photons and an area detector to measure the profile of the scattered beam from the analyzer. The input beam is monochromatic and collimated. The samples we used are single crystals GaAs and LiNbO$_3$. We tuned the angles of the sample and the detector to the phase-matching angles of the selected reciprocal lattice vector and photon energies. We rocked the sample and recorded the signal with the detector. Typical images of Bragg reflection and of the PDC signal are shown in Fig. 1c, d, respectively. The Bragg reflection and the PDC signal are distinctive and are spatially separated from each other. From the images we construct the rocking curves of the sample (the signal count rate as a function of the sample angle) for various signal photon energies. At each of the energies we verify that we measure the PDC signal by comparing the positions of the peaks of the rocking curves to the calculated phase-matching angles. We then reconstruct the spectra by registering the peak values of the rocking curves at each of the energies. Examples for the rocking curves can be found in the Supplementary Note 1 and Supplementary Figs. 1–4.

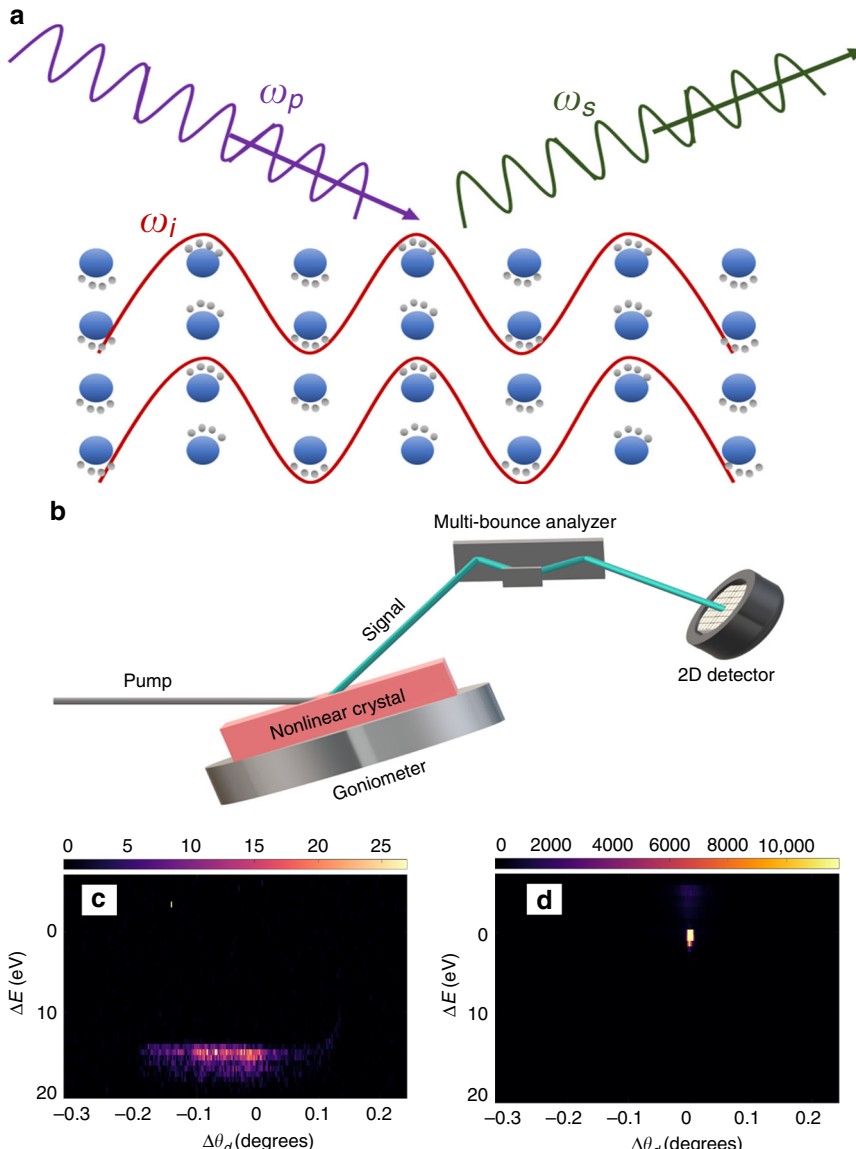

**Fig. 1 Schematic description of the physical process and of the experiment. a** Description of the mechanism for the nonlinearity. The pump X-ray beam is scattered by an optically modulated charges and the frequency of the scattered X-rays is redshifted. **b** The experimental setup. The synchrotron radiation illuminates a nonlinear crystal. The PDC signal is selected by a multi-bounce analyzer and measured by a two-dimensional detector. The vertical axis of the detector is the deviation of the detected energy from the pump energy, which is determined by the analyzer angle. The horizontal axis is the angle with respect to the Bragg angle that corresponds to the pixel on the camera. **c** Typical image of Bragg reflection as taken by the detector. **d** Typical image of PDC. Note that while the profile of the Bragg reflection is very narrow in the horizontal direction the PDC profile is broad.

**Observation of strong nonlinear interactions**. We begin by showing the very strong efficiencies of the signal of the PDC of X-rays into UV in both GaAs and LiNbO$_3$. Here we define the efficiency as the sum over the full-width at half-maximum of the PDC signal, as recorded by the detector, divided by the incident flux, while the sample is at the phase-matching angle. The data analysis process is described schematically in the Supplementary Note 2 and in Supplementary Fig. 5. We plot the efficiencies of the PDC for various idler energies and atomic planes in the GaAs and in the LiNbO$_3$ crystals in Figs. 2 and 3, respectively. In the figures, the abscissa is the interplanar spacing and the vertical axis is the measured efficiency. The efficiencies range from $10^{-8}$ to $10^{-6}$, where in previous measurements of X-ray into UV PDC the efficiencies were on the order of $10^{-9}$–$10^{-10}$ (refs. [13,14]). The general trend far from atomic resonances is that the efficiency drops, as the idler photon energy increases. This is mainly because

the current density, which is proportional to the joint density of states, decreases when the idler photon energy increases[25]. In addition, the theory, which led to a good agreement in the previous experiments[13–15], underestimates the efficiencies for GaAs and LiNbO$_3$ by about four orders of magnitude. This ratio is maintained in the whole range of measured photon energies. These pronounced discrepancies indicate on a new source for the strong nonlinearity, which was not considered in previous works.

Of importance, the measured efficiencies presented in Figs. 2 and 3 show a very different angular dependence from the angular dependence of the elastic scattering. Hence our results cannot be described by generalized scattering factors. The interpretation of the measured efficiencies for different directions of the crystal is essential for the understanding of the nonlinear interaction. The differences in the efficiencies for different atomic planes can be

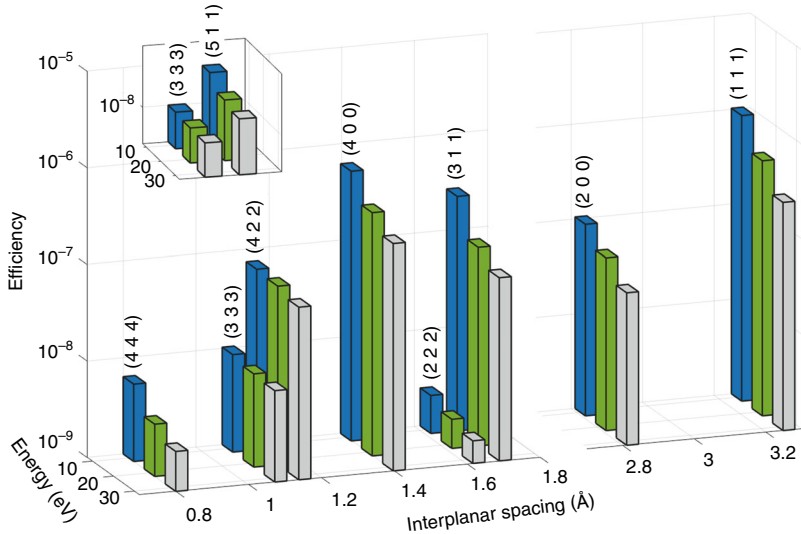

**Fig. 2 Comparison between measured efficiencies of PDC in GaAs.** Several atomic planes are shown for a bandwidth of 1 eV. The presented efficiencies are measured for signals, which correspond to idler energies of 10, 20, and 30 eV. The inset presents the efficiencies measured for two atomic planes with the same interplanar spacing. The horizontal axis represents the interplanar spacing, which corresponds to the Miller indices of the various atomic planes.

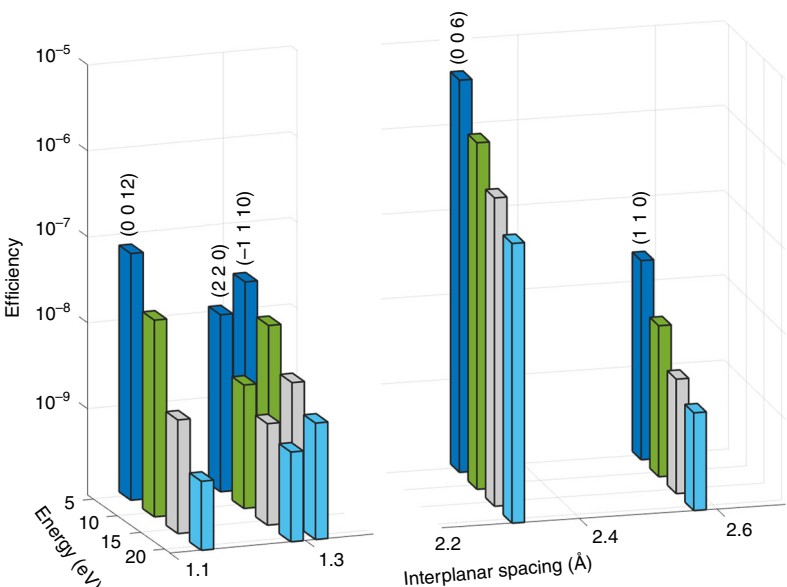

**Fig. 3 Comparison between measured efficiencies of PDC in LiNbO₃.** Several atomic planes are shown for a bandwidth of 1 eV. The presented efficiencies are measured for signals, which correspond to idler energies of 5, 10, 15, and 20 eV. The horizontal axis represents the interplanar spacing, which corresponds Miller indices of the various atomic planes.

attributed to higher densities of valence electrons in different directions.

Another interesting result becomes apparent when comparing the efficiencies for LiNbO₃ with reciprocal lattice vectors of nearly equal magnitudes. Although the X-ray scattering factors for these reciprocal lattice vectors are comparable, the efficiencies of the PDC effect for reciprocal lattice vectors, which are parallel to the *c*-axis of the crystal, are considerably larger. For example, the effect measured for the (0 0 6) atomic planes is two orders of magnitude larger compared to the (1 1 0) atomic planes. This can indicate that the nonlinearity depends on the direction of the permanent dipoles in the crystal and on symmetry of unit cells or the molecules in the materials. It is remarkable to note that for the visible range, the nonlinear coefficient is the largest for polarizations along the *c*-axis[27]. It is very likely that the electric field of the idler that leads to the largest efficiencies we observed

in LiNbO₃ is also in the direction of the *c*-axis. This is consistent with the theories that predict that the largest efficiency for selected atomic planes is obtained when the electric field of the idler is parallel to the reciprocal lattice vector (normal to the atomic planes)[15].

**Spectral measurements and spectroscopy applications.** After we show the observation of the very strong nonlinearities, we elucidate the origin of the effect and demonstrate the ability of its application for broad range spectroscopic measurements by exploring the spectral dependencies of the effect that are shown in Figs. 4 and 5 with a higher photon energy resolution. Figure 4a shows the spectrum measured for the GaAs (2 0 0) atomic planes. Here we show four clear features at idler energies of 8 eV, 17 eV, 28 eV, and 38 eV. We interpret the first two peaks as either direct

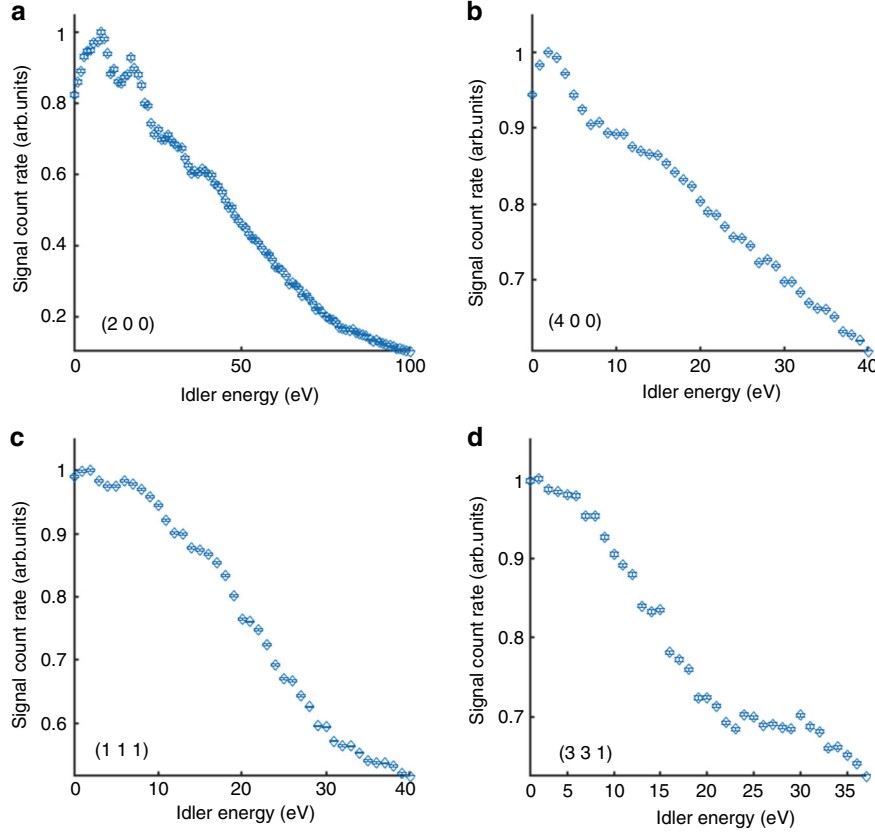

**Fig. 4 Spectral measurement of the PDC signal in GaAs.** Spectra for **a** the (2 0 0) atomic planes, **b** the (4 0 0) atomic planes, **c** the (111) atomic planes, and **d** the (3 1 1) atomic planes. The features in **a** and **b** can be attributed to atomic resonances or to inter-band transitions in the crystal. The vertical error bars indicate the counting statistics. See further details in the text.

transitions related to the band structure of GaAs or atomic resonances of the arsenic atoms. The other two peaks cannot be attributed to band structure transitions (according to the present literature on the calculations of the band structure of GaAs[28]), but can be attributed to deeper atomic resonances of the arsenic atoms. The spectrum of the GaAs (4 0 0) atomic planes is shown in Fig. 4b. Although this plane is parallel to the GaAs (2 0 0) planes, the measured spectra are clearly very different. This difference can be explained by considering the zinc-blende crystal structure of the GaAs, which suggests that the atoms in the GaAs (2 0 0) atomic planes are either gallium or arsenic atoms. This explanation is supported by observations in Fig. 4a that show only atomic resonance of arsenic atoms. The peak that we observe for the GaAs (4 0 0) planes is interpreted as the transition of the direct band gap of GaAs. The measured spectra shown in Fig. 4c and d correspond to the GaAs (1 1 1) and GaAs (3 1 1) atomic planes, respectively. The spectra are still non-monotonic in the energy dependencies, with features that are above the noise level, but are less pronounced than the features shown in the previous spectra. The measured spectrum of the GaAs (1 1 1) atomic planes indicates that the highest efficiency is slightly above the band gap of GaAs as was measured for the (4 0 0) reflection, in addition to a feature at 6 eV, which can correspond to either transition at the band structure or to an ionization energy for the 4$p$ valance electrons in gallium atoms. The spectrum measured for the GaAs (3 1 1) atomic planes shows again a maximal efficiency slightly above the band gap. In addition, we observe a sharp change in the trend of the spectrum near 20 eV. This feature is different than those discussed previously and cannot be interpreted as a resonance (from the band structure or atomic transitions).

In Fig. 5 the measured spectra of the LiNbO$_3$ crystal are shown. The peak at 5.9 eV for LiNbO$_3$ (1 1 0) and LiNbO$_3$ (3 3 0) atomic planes and the broad structures we observed for the LiNbO$_3$ (−1 1 10) atomic planes, which are shown in Fig. 5a–c, respectively, can be attributed either to the direct band transition[29] or to the ionization energy for the 2$s$ valance electrons in lithium atoms. The peak at 7.5 eV, which appears in Fig. 3a, b, can be attributed either to the transition between the valence and the second conductance bands or to the ionization energy for the 4$s$ electrons in niobium atoms. The peaks at 12 and 15 eV, which appears for the LiNbO$_3$ (−1 1 10) and the LiNbO$_3$ (3 3 0) atomic planes relate either to higher band transitions or to a deeper atomic level. Figure 5d shows the spectrum for the LiNbO$_3$ (0 0 6) atomic planes. In these measurements the atomic planes that participate in the effect are directed along the polarized direction on the crystal. This spectrum shows no prominent features.

The main implication of the measured spectra is that the effect of PDC of X-ray into longer wavelength radiation can be used as a powerful tool to investigate phenomena in solid-state physics and in atomic physics. Generally speaking, it is likely that the lower idler energies correspond to solid-state physics and the higher energies to atomic physics, but to exactly distinguish these effects, a more detailed theoretical model is needed. Actually, the transition between the range where the dominant physical mechanism is solid-state physics and the range where the atomic physics dominates has been rarely studied. Hence, we foresee that PDC of X-rays into UV will be used as a powerful tool for studying this range, which will open an opportunity to study a broad range of physical phenomena with one experimental setup. We note that the peaks at the band gap of the crystal that appear in most of the measurements can be explained by the divergence

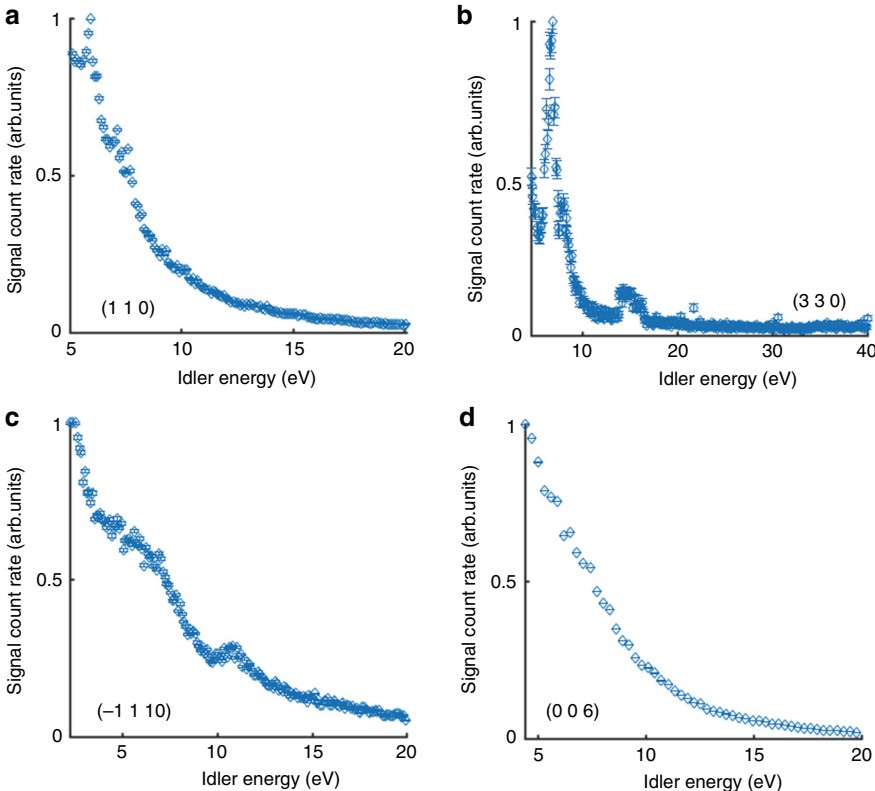

**Fig. 5 Spectral measurements for PDC signal in LiNbO₃.** Spectra for **a** the (1 1 0) atomic planes, **b** the (3 3 0) atomic planes, **c** the (−1 1 10) atomic planes, and **d** the (0 0 6) atomic planes. The features in **a**–**c** can be attributed to either band transitions or atomic resonances. The vertical error bars indicate the counting statistics. See further details in the text.

of the joint density of states[25]. This suggests that PDC of X-rays into UV can be used for the study of electronic density of states and band structures in solids.

Another pronounced property that is common to all spectra is that apart from local enhancement near resonances, the efficiency decreases with the idler photon energies. However, the efficiency spectra are very different and depend on the atomic planes that are used for the phase matching. In addition, the efficiency dependencies on the idler energy in LiNbO₃ are stronger than in GaAs. This can be explained by noting that in LiNbO₃ there are fewer atomic resonances in the measured energy range. The different visible features in each of spectra can be explained by the nonlinearity, which depends also on the reciprocal lattice vectors themselves and on the distributions of the electronic wave functions. A comprehensive understanding of the spectra should include analysis of the nonlinear conductivity for different reciprocal lattice vectors while considering the selection rules between transitions.

**Dependence on inversion symmetry**. We note that both inspected crystals have non-centrosymmetric crystal structures. This is in contrast to silicon and diamond crystals that have been investigated before. This observation raises the question of the importance of inversion symmetry to nonlinear effects with X-rays, which has been ignored in all previous publications. It is well known that in the optical regime the lack of inversion symmetry is essential for second-order nonlinearities; however, thus far, X-ray nonlinearities have been considered as resulting from inter-actions beyond the diploe approximation that can be viewed as X-ray scattering from optically driven oscillations[11]. This type of second-order nonlinearity is non-zero even in centrosymmetric martials but significantly weaker than the ordinary second-order

nonlinearity in non-centrosymmetric materials in the optical regime. It is therefore possible that the lack of inversion symmetry contributes significantly to the strong nonlinear effects we report here, which mixes between X-rays and long wavelengths. While a comprehensive model that describes the interaction that we have measured in our experiments is still not available, we can test this hypothesis experimentally by comparing the PDC signal in the non-centrosymmetric and in centrosymmetric phases of the same crystal. A good example for such a crystal would be a ferroelectric crystal such as LiNbO₃ since the ferroelectric phase is known to be non-centrosymmetric whereas the paraelectric phase is centrosymmetric. Since the Curie temperature ($T_c$) of LiNbO₃ is 1145 °C, we choose to explore the dependence on the inversion symmetry in $Sr_{0.6}Ba_{0.4}Nb_2O_6$ crystals, where $T_c$ is between 340 and 350 K[30], thus more accessible with conventional heaters. We performed our measurements on beamline I16 of the Diamond Light Source[26], with a setup, which is similar to that in Fig. 1b with the addition of a nitrogen gas jet, which we used to control the temperature of the crystal. At temperatures below the $T_c$, SBN is ferroelectric and has a tetragonal non-centrosymmetric crystal structure. Above $T_c$, the structure of SBN is cubic. Although the crystal is not completely centrosymmetric in this phase, due to the presence of various types of atoms in its unit cell, the degree of the inversion symmetry is much higher compared to the ferroelectric phase.

The angular dependence of the PDC signal in both phases is shown in Fig. 6. The results are very clear; the measured PDC signal in the ferroelectric phase is about two orders of magnitude higher than the efficiency in the paraelectric phase. These results support the hypothesis that the lack of inversion symmetry is the physical origin of strong nonlinearity we observed in non-centrosymmetric crystals.

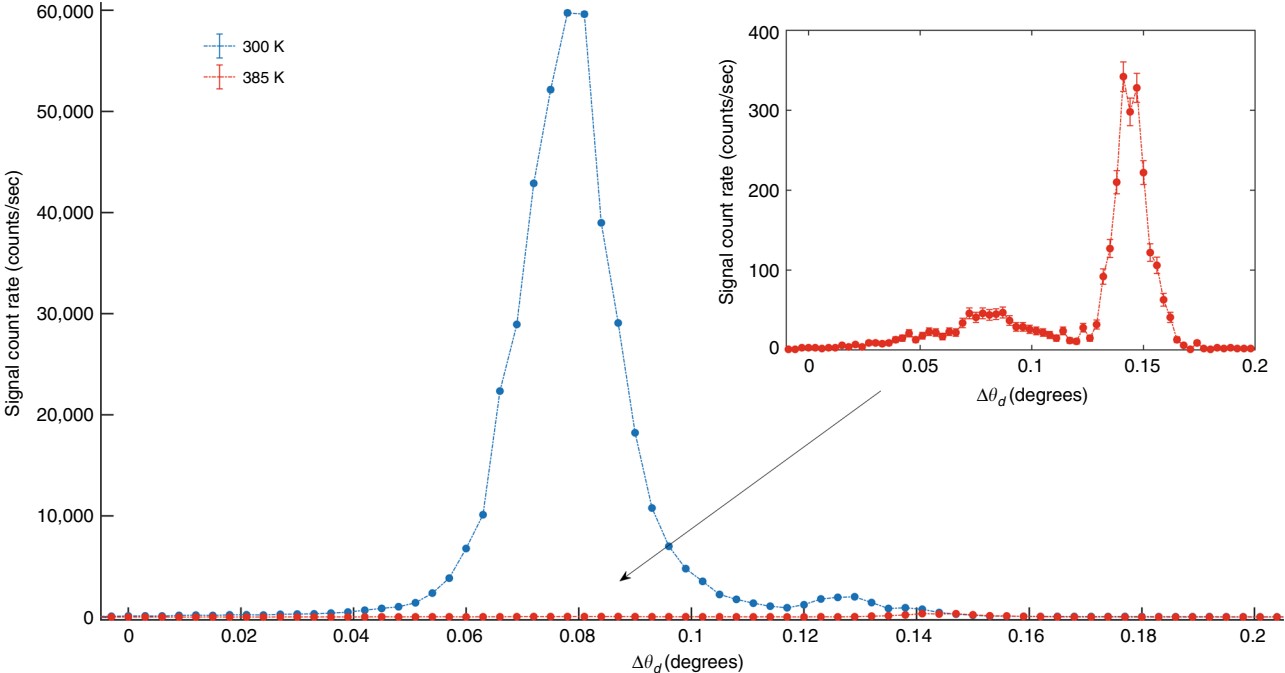

**Fig. 6 Comparison between PDC in the ferroelectric and the paraelectric phases of SBN.** The angular dependencies of the PDC signal corresponds to idler energy at 20 eV, for the tetragonal phase at 300 K (blue dots), and cubic phase at 385 K (red dots), for the (2 1 3) atomic planes. The two peaks in each curve correspond to the two solutions of the phase-matching equation. The Curie temperature of the crystal is between 340 and 350 K[30]. The horizontal axis is the signal angle with respect to the Bragg angle. The inset shows the signal angular dependence at the cubic phase. The vertical error bars indicate the counting statistics.

In conclusion, we have reported the measurement of non-predicted strong nonlinearities leading to efficiencies that are larger by about four orders of magnitude than previously reported efficiencies of PDC of X-rays into long wavelength radiation. These high efficiencies cannot be explained by existing theories and indicate an unexplored physical underlying mechanism. We note that the angular dependence of the effect, the dependence on the pump photon energy, and the agreement with the phase-matching calculations constitute strong evidence that the effect we have measured is indeed PDC and not any other known inelastic effect. Furthermore, we have found that at the edges the PDC efficiency is reduced in contrast to inelastic scattering effects[31]. We have provided evidence that the lack of inversion symmetry is the physical mechanism for the strong nonlinearity in non-centrosymmetric crystals. Consequently, it is likely that our method can be applied to a large class of non-centrosymmetric materials and will lead to the development of advanced tools for the investigation of inversion symmetry breaking mechanisms, anharmonicity, and interfaces on the atomic scale. Our work demonstrates the possibility to utilize the effect of PDC of X-rays into visible and UV radiation as a powerful tool to probe numerous physical phenomena as it covers a large range of energies and provides structural and spectroscopic information in a single measurement. We expect that the full understanding of the effect will open a large range of opportunities to probe properties of solids and atoms, which are currently obscured due to the limitations of the present-day methods, hence advance the understanding of their functionality. Our work can be extended into the studying of the dynamics of the valence electrons and electronic excitations in a broad spectral range. This can be done by combining the X-ray input beam for the PDC with an additional optical laser in a pump-probe configuration. These types of experiments will be of interest with X-ray free-electron lasers, which provides femtosecond pulses as compared to picosecond pulses available with storage ring-based

synchrotrons. We expect that the data acquisition time with the new emerging high repetition-rate free-electron lasers will be content[32].

## Methods

**Experimental setup**. We provide further information on the experimental setups, which we used for both the GaAs and the LiNbO₃ experiments. First, we describe the experimental setup for the GaAs experiment, which we performed at Diamond Light Source on beamline I16. The input power is approximately $10^{13}$ photons/s. The input beam is monochromatic at 10.3 keV and is polarized in the scattering plane. The dimensions of the beam at the input are 20 μm (vertical) × 180 μm (horizontal). The nonlinear crystal is a GaAs crystal with a thickness of 350 μm. The analyzer is an Si (1 1 1) triple-bounce analyzer and the detector is a Medipix multipixel detector with pixel size of 55 μm × 55 μm. The overall spectral resolution of the system is about 1 eV. The experimental setup, which we use for the SBN measurements, is similar to that of the GaAs. A minor difference is the input energy, which is at 10 keV. Additionally, we use a nitrogen gas jet heater to heat the crystal.

We performed the LiNbO₃ measurements at the ESRF on the ID20 beamline with a similar setup. The input power is approximately $5 \times 10^{12}$ photons/s. The input beam is monochromatic at 10 keV and is polarized in the scattering plane. The dimensions of the beam at the input are 0.4 mm × 0.4 mm. The nonlinear crystal is a polished LiNbO₃ crystal with dimensions of 7 mm × 7 mm × 5 mm. The analyzer is an Si (4 4 0) double-bounce analyzer and the detector is a Medipix multipixel detector with a pixel size of 55 μm × 55 μm. The overall spectral resolution of the system is about 0.3 eV.

In all of the experiments the scattering plane of the analyzer was perpendicular to the scattering plane of the nonlinear crystal, in order to decouple the analyzer angle from the signal angle.

**Analysis procedure**. We add further details on the data analysis procedure. We select the photon energy by tuning the analyzer angle. At each of the photon energies we scan the angle of the sample and record the intensity on the area detector for exposure times between 0.01 and 1 s. At each angle of the sample we sum over the counts in the area within full-width at half-maximum in the horizontal axis of the detector. We then plot the rocking curve (the dependence of the efficiency on the angle of the sample) and find its peak value. We repeat this procedure for the entire range of photon energies and plot the dependence of the peak value of the rocking curve as a function the photon energy to construct the spectra that we plot in Figs. 4 and 5.

To calibrate the vertical pixels of the detector with respect to the idler energy, we use the following procedure. We tune the sample and the detector arm to the Bragg angles and the crystal analyzer to observe the highest intensity. This condition corresponds to the elastic scattering. We then find the vertical position on the detector where the Bragg signal is observed. By using the energy conservation equation $\omega_p = \omega_s + \omega_i$, we determine that this position corresponds to the idler energy of 0 eV, i.e. the elastic energy. Next, we detune the analyzer angle to fit for signal energy of 15 eV off the elastic energy. Energy conservation determines that this signal corresponds to idler energy of 15. We use linear interpolation to determine the idler energy along the vertical direction of the detector. The transformation of the horizontal position along the detector into degrees is performed as follows. We measure the Bragg horizontal position. This position is set to be the calculated Bragg angle. Next, we calculate the deviation from the Bragg angle by dividing the horizontal distance along the detector by the distance of the detector from the crystal. We repeat the calibration procedure for each of the atomic planes.

## Data availability

The data that support the findings of this study are available from the corresponding author upon request.

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

## Acknowledgements

The authors thank Dr. Aviad Schori for provision of data acquisition and analysis programs. We acknowledge Prof. Nimrod Moiseyev for helpful discussions. We acknowledge the European Synchrotron Radiation Facility for provision of synchrotron radiation facilities. We acknowledge Diamond Light Source for time on Beamline I16 under Proposals [MT17527] and [MM22041]. This work was supported by the Israel Science Foundation (ISF) (IL), Grant No. 201/17. The research leading to this result has been supported by the project CALIPSOplus under Grant Agreement 730872 from the EU Framework Programme for Research and Innovation HORIZON 2020.

## Author contributions

O.S., S. Sofer, and S. Shwartz conceived the experiments. S. Sofer, O.S., E.S., H.A., B.D., S. P.C., G.N. Ch.J.S., and S. Shwartz collected the data. S. Sofer and O.S. preformed the data analysis. B.D., S.P.C., G.N., and Ch.J.S. contributed to the experimental design and installation. S. Shwartz supervised the project. S. Sofer, O.S., and S. Shwartz wrote the manuscript. All authors contributed to the work presented here and to the final paper.

## Competing interests

The authors declare no competing interests.
