## [Peer Review File · Nature Communications]

Reviewers' Comments:

Reviewer #1:

Remarks to the Author:

The paper reports a significant results of X-ray nonlinear phenomena that non-centrosymmetric crystals give an enhanced PDC signals. The previous reports dealt only with the centrosymmetric crystals which do not show such enhanced PDC signals. In addition, the paper deals with the lattice plane dependence of the PDC signals that is also new.

The new observations is challenging our understanding of the nonlinear x-ray optics. But, unfortunately, the paper does not mention to the fundamental physics making leading to the significant results but says something very general.

I would like suggest authors to elaborate a little more on the basic idea which explains the enhancement of the PDC signals from non-centrosymmetric crystals. Without it, I cannot agree that the paper will be published.

Reviewer #2:

Remarks to the Author:

The manuscript entitled "Observation of strong nonlinear interactions in parametric down conversion of x-rays into ultraviolet radiation" by S. Sofer and co-workers reports parametric down-conversion (PDC) experiments from GaAs and LiNbO₃ samples. The data show a PDC efficiency largely above the range typically reported for other materials.

The development of nonlinear x-ray spectroscopies has attracted much attention on the last years, following several theoretical papers (e.g. from the Mukamel's group) and, most recently, experiments based on x-ray lasers carried out by different experimental teams worldwide. PDC and its stimulated analogue (sum or difference frequency generation) are nonlinear processes that has can be exploited for building up x-ray spectroscopies (refs 8-18). In such a context the experimental data reported in the present work are valuable and worth publication. However, I believe that at the present stage the interpretation of the observed phenomenology is not sufficient for supporting the central claim of the manuscript, i.e. that these data demonstrate how PDC can be used to probe a broad several phenomena in materials.

More specifically, the perspective of using the PDC as a tool for combining the valence-band sensitivity of optical photons with the atomic-scale spatial resolution and chemical selectivity of x-ray photons has been already presented in the literature; these results are properly cited in the manuscript. The main additional finding showed in this work is the very large increase in the PDC efficiency in two given materials, and how this cannot be accounted for by conventional theories. The authors soundly describe the observed spectral features, which, I guess, are made observable by the large nonlinear signal, and provide a quantitative description of the PDC intensity vs the experimental parameters (e.g. the sample orientation). However, the central point (i.e. the origin of this large nonlinear signal) is not addressed and, at the end of the story, the underlying physical mechanisms remain unknown. This is somehow recognized by the authors in the discussion section ("We expect that the full understanding of the effect will open..."). In summary, such a lack of knowledge prevents generalizing these findings to all materials, or to some classes of materials, thus making not convincing the aforementioned claim.

Therefore, I do not recommend publication in Nature Communications and I suggest to bring this work to the attention of a more specialized journal.

I'd also like to give my personal view on a possible outlook for this study. In my opinion the most likely reason for the very large nonlinear signal should be related to the non-centrosymmetric structure, as mentioned in the text; the observation of a larger signal along the c-axis also goes towards this direction. A glassy clear experimental proof for that (I'm thinking, for example, to comparative PDC measurements on the centrosymmetric and non-centrosymmetric phases of different samples), even without a comprehensive theoretical explanation, would be sufficient for

claiming that PDC can be exploited in a much more general context. Namely for probing any non-centrosymmetric sample, as well as any surface/interface, where, obviously, the inversion symmetry is broken.

Reviewer #3:

Remarks to the Author:

The authors have observed strong x-ray down conversion at two synchrotron facilities, Diamond and ESRF, which they interpret as nonlinear parametric down conversion, with some unanswered questions regarding the nature of the idler. X-ray nonlinear studies are of great interest as a new scientific tool at both synchrotron and free electron laser (FEL) facilities, particularly with opportunities for femtosecond and attosecond studies at the XFELs. The authors results are intriguing and are likely to provoke further studies.

Observation of strong nonlinear interactions in parametric down-conversion of x-rays into ultraviolet radiation

S. Sofer, O. Sefi, E. Strizhevsky, H. Aknin, S.P. Collins, G. Nisbet, B. Detlefs, Ch.J. Sahle, and S. Shwartz

Detailed response to the comments of the Reviewers

A detailed response to the Reviewer 1

Reviewer 1 recommends that we elaborate on the physical mechanism that supports the enhancement of the PDC signals in non-centrosymmetric crystals.

Our reply: We thank the reviewer for this important comment and agree that this point has to be addressed. In accordance with this comment and the main comment of Reviewer 2 we have conducted another experiment that address this question. We study the role that inversion symmetry plays in the nonlinear conversion of x-rays into UV by measuring PDC of x-rays into UV in another crystal ($\text{Sr}_{0.6}\text{Ba}_{0.4}\text{Nb}_2\text{O}_6$), which is ferroelectric at room temperature and becomes paraelectric between 340 K to 350 K. In the ferroelectric phase the crystal is non-centrosymmetric and in the paraelectric the crystal is centrosymmetric. Our measures reveal that the PDC effect is stronger by nearly two orders of magnitude in the non-centrosymmetric (ferroelectric) phase. This result strongly supports the hypothesis that the anharmonicity is dominant origin of the enhancement of the nonlinearity we observed. We have added the new results and a discussion in page 16. We hope that the reviewer will find the new supporting evidence and discussion satisfactory and recommend on publishing our manuscript in Nature Communications.

A detailed response to Reviewer 2

The main comment of Reviewer 2 is “In summary, such a lack of knowledge prevents generalizing these findings to all materials, or to some classes of materials, thus making not convincing the aforementioned claim.

Therefore, I do not recommend publication in Nature Communications and I suggest bringing this work to the attention of a more specialized journal.

I’d also like to give my personal view on a possible outlook for this study. In my opinion the most likely reason for the very large nonlinear signal should be related to the non-centrosymmetric structure, as mentioned in the text; the observation of a larger signal along the c-axis also goes towards this direction. A glassy clear experimental proof for that (I’m thinking, for example, to comparative PDC measurements on the centrosymmetric and non-centrosymmetric phases of different samples), even without a comprehensive theoretical explanation, would be sufficient for claiming that PDC can be exploited in a much more general context. Namely for probing any non-centrosymmetric sample, as well as any surface/interface, where, obviously, the inversion symmetry is broken.”

Our reply: We thank the reviewer for reviewing our manuscript and for proposing a new experiment to explore the origin of the string observed enhancement. We agree that in the original manuscript we did not provide any potential explanation for our observations. As in our reply to Reviewer 1, we have performed exactly the experiment reviewer 2 suggested. We have measured PDC of x-ray into UV in $\text{Sr}_{0.6}\text{Ba}_{0.4}\text{Nb}_2\text{O}_6$ crystal, which is ferroelectric at room temperature (non-centrosymmetric) and paraelectric above 350 K (centrosymmetric). As anticipated, the efficiency we have measured in the ferroelectric phase (the non-centrosymmetric) is comparable to the efficiency in the other non-centrosymmetric crystals but the efficiency in the paraelectric phase (centrosymmetric) is about two orders of magnitude smaller. This result indicates that the noncentrosymmetry is the physical mechanism for the nonlinearity we observed. It suggests that our method can be exploited in a general context of other non-centrosymmetric crystals and show that our results have broad implications to many other materials. We have added the new results and elaborate on their implications in the text (page 16). We hope that the new information and the discussion we have added are sufficient to convince the reviewer to recommend on accepting our manuscript for publication in Nature Communications.

A detailed response to Reviewer 3

Reviewer 3 did not raise any particular concern.

Reviewers' Comments:

Reviewer #1:

Remarks to the Author:

The authors have shown the enhancement of the PDC effect by using non-centrosymmetric samples. This is a new and interesting finding. Although the detailed discussion on the origin of enhancement was not clearly given, I judged the previous version of the manuscript being above the boundary of acceptance. This revised version, with the helps of the other referees, has been improved so much that I can judge it well above the boundary of acceptance.

Therefore, I DO support the publication in Nature Communication.

Reviewer #2:

Remarks to the Author:

I'm glad to see that the authors seriously considered my suggestion, up to point of performing an additional experiment.

According to the new data, I'm still doubtful regarding the perspective of using PDC as an experimental tool to probe any material. The orders of magnitude decrease in the signal intensity from paraelectric SBN suggests that in centrosymmetric samples the level of details cannot be comparable with those shown in Figs 4-5. However, the contrast in the experimental signal from centrosymmetric vs non-centrosymmetric structures is a really important result! I mean, the authors are demonstrating a method with symmetry breaking sensitivity using an inherently bulk probe with atomic-scale resolution. On such grounds one can envision a number of exciting applications, for instance spectroscopic and structural studies of buried interfaces (this class of samples includes a number of technologically relevant systems that to date can be studied only with long wavelength probes), biological membranes or protein surfaces, just to say a few examples that immediately come up in my mind. I strongly suggest to revise some statements throughout the paper in order to better focus the range of applications. For instance, the statement (in the abstract): "...investigate the spectral response of materials in a broad..." should be "...investigate the spectral response of non-centrosymmetric materials, surfaces and interfaces in a broad..." or something similar. Also in the conclusions one should specify what they mean with "large class of materials", highlighting the main ones. This will enhance the relevance of the results, by directly addressing the scientific communities that can better exploit such an approach.

The authors mention that "The general trend far from atomic resonances is that the efficiency drops as the idler photon energy increases". This point could be further elaborated. The general decreasing trend is due to a decrease in the (non-resonant) non-linear susceptibility vs the idler frequency? Perhaps it could be described by a Lorentian shape? (I would expect that by looking at the classical trend of non-resonant non-linear optical susceptibility). These data represent relevant info on the nonlinear response at high photon frequency and, if possible, deserve a bit of interpretation; at least they must be properly presented for what they are and their relevance should be stated.

In the conclusions the author worthy mention FEL experiments. In this context it's also worth a brief discussion on the different experimental conditions. For PDC the much higher repetition rate of a synchrotron is likely more beneficial than the much higher peak brightness of an FEL, therefore high repetition rate FELs (currently in realization) are probably needed for advancing the PDC approach. Besides PDC, FELs also allow stimulated processes (e.g. SHG or SFG; see Ref. 11 and PRL 112, 163901 (2014)), which should be mentioned in the conclusions.

In general I found the narration too emphatic, a plain narration avoiding emphatic expressions should be adopted. Also, authors use very often the term "long wavelengths" for the idler, "long

wavelength photons" or "long wavelength radiation" is a more appropriate wording.

In conclusion, the new data have addressed my previous concern, as a consequence I've changed my mind and I recommend publication in Nature Communications (and I hope that the above suggestions can help in improving the manuscript).

Observation of strong nonlinear interactions in parametric down-conversion of x-rays into ultraviolet radiation

S. Sofer, O. Sefi, E. Strizhevsky, H. Akin, S.P. Collins, G. Nisbet, B. Detlefs, Ch.J. Sahle, and S. Shwartz

Detailed response to the comments of the Reviewers

A detailed response to the Reviewer 1

Reviewer 1 considers the manuscript above the acceptance boundary and recommends on its publication in Nature Communications.

Our reply: We thank the reviewer for reviewing our manuscript and for recommending on the acceptance of our manuscript for the publication in Nature Communications.

A detailed response to Reviewer 2

After we have added the new experimental results and the pertinent discussion, Reviewer 2 has changed his/her recommendation and now the reviewer recommends publication of our manuscript in Nature Communications.

However, the reviewer made several comments to improve the readability of the manuscript. We have addressed each of the points the reviewer raised and describe our detailed response below.

Comment 1: " According to the new data, I'm still doubtful regarding the perspective of using PDC as an experimental tool to probe any material. The orders of magnitude decrease in the signal intensity from paraelectric SBN suggests that in centrosymmetric samples the level of details cannot be comparable with those shown in Figs 4-5. However, the contrast in the experimental signal from centrosymmetric vs non-centrosymmetric structures is a really important result! I mean, the authors are demonstrating a method with symmetry breaking sensitivity using an inherently bulk probe with atomic-scale resolution. On such grounds one can envision a number of exciting applications, for instance spectroscopic and structural studies of buried interfaces (this class of samples includes a number of technologically relevant systems that to date can be studied only with long wavelength probes), biological membranes or protein surfaces, just to say a few examples that immediately come up in my mind.

I strongly suggest to revise some statements throughout the paper in order to better focus the range of applications. For instance, the statement (in the abstract): "...investigate the spectral response of materials in a broad..." should be "...investigate the spectral response of non-centrosymmetric materials, surfaces and interfaces in a broad..." or something similar. Also in the conclusions one should specify what they mean with "large class of materials", highlighting the main ones. This will enhance the relevance of the results, by directly addressing the scientific communities that can better exploit such an approach."

Our replay: This is a very good point. We have revised the statements throughout the manuscript according to the suggestions of reviewer and emphasize that the implications of our manuscript are important for non-centrosymmetric materials.

Comment 2: "The authors mention that "The general trend far from atomic resonances is that the efficiency drops as the idler photon energy increases". This point could be further elaborated. The general decreasing trend is due to a decrease in the (non-resonant) non-linear susceptibility vs the idler frequency? Perhaps it could be described by a Lorentian shape? (I would expect that by looking at the classical trend of non-resonant non-linear optical susceptibility). These data represent relevant info on the nonlinear response at high photon frequency and, if possible, deserve a bit of interpretation; at least they must be properly presented for what they are and their relevance should be stated."

Our replay: We agree that this point should be explained further. We have added the sentence "The general trend far from atomic resonances is that the efficiency drops, as the idler photon energy increases. This is mainly because the current density, which is

proportional to the joint density of states, decreases when the idler photon energy increases " and a pertinent reference to the discussion on figures 2 and 3 on page 7 of the main text.

Comment 3: " In the conclusions the author worthy mention FEL experiments. In this context it's also worth a brief discussion on the different experimental conditions. For PDC the much higher repetition rate of a synchrotron is likely more beneficial than the much higher peak brightness of an FEL, therefore high repetition rate FELs (currently in realization) are probably needed for advancing the PDC approach. Besides PDC, FELs also allow stimulated processes (e.g. SHG or SFG; see Ref. 11 and PRL 112, 163901 (2014)), which should be mentioned in the conclusions."

Our replay: We have added a brief discussion about the differences between XFEL and storage ring-based synchrotrons. Additionally, we have elaborated more about the possibility to use this technique for the study of the dynamics of the valence electrons in crystals and added the following text "Our work can be extended into the studying of the dynamics of the valence electrons and electronic excitations in a broad spectral range. This can be done by combining the x-ray input beam for the PDC with an additional optical laser in a pump-probe configuration. These types of experiments will be of interest with x-ray free-electron lasers, which provides femtosecond pulses as compared to picosecond pulses available with storage ring based synchrotrons. We expect that the data acquisition time with the new emerging high repetition-rate free-electron lasers will be content" at the end of the main text.

Comment 4: " In general I found the narration too emphatic, a plain narration avoiding emphatic expressions should be adopted. Also, authors use very often the term "long wavelengths" for the idler, "long wavelength photons" or "long wavelength radiation" is a more appropriate wording."

Our replay: As Reviewer 2 suggested, we have revised the term "long wavelengths" to "long wavelength radiation" throughout the manuscript.

Summary: We sincerely thank Reviewer 2 for carefully reviewing our manuscript and for making valuable comments that improve the readability of our manuscript. We have revised the manuscript according to the comments of the reviewer and believe that the current version of the manuscript is significantly improved.